# Demonstration of Accuracy and Feasibility of Remotely Delivered Oximetry: A Blinded, Controlled, Real-World Study of Regional/Rural Children with Obstructive Sleep Apnoea

**DOI:** 10.3390/healthcare11020278

**Published:** 2023-01-16

**Authors:** Ryan Begley, Yogesan Kanagasingam, Chun Chan, Chandrashan Perera, Moya Vandeleur, Paul Paddle

**Affiliations:** 1Department of Otolaryngology Head & Neck Surgery, Monash Health, Melbourne, VIC 3168, Australia; 2School of Medicine, University of Notre Dame, Fremantle, WA 6160, Australia; 3Department of Ophthalmology, WA Health, Perth, WA 6000, Australia; 4Department of Respiratory & Sleep Medicine, The Royal Children’s Hospital Melbourne, Melbourne, VIC 3052, Australia; 5Department of Surgery, Faculty of Medicine, Nursing & Health Sciences, Monash University, Melbourne, VIC 3800, Australia

**Keywords:** digital health, remote diagnostics, remote care, paediatrics, surgical assessment, rural and regional patients

## Abstract

Objectives: Evaluate diagnostic accuracy and feasibility of a mail-out home oximetry kit. Design: Patients were referred for both the tertiary/quaternary-centre hospital-delivered oximetry (HDO) and for the mail-out remotely-delivered oximetry (RDO). Quantitative and qualitative data were collected. The COVID-19 pandemic began during this study; therefore, necessary methodological adjustments were implemented. Setting: Patients were first evaluated in Swan Hill, Victoria. RDO kits were sent to home addresses. For the HDO, patients travelled to the Melbourne city area, received the kit, stayed overnight, and returned the kit the following morning. Participants: All consecutive paediatric patients (aged 2–18), diagnosed by a specialist in Swan Hill with obstructive sleep apnoea (OSA) on history/examination, and booked for tonsillectomy +/− adenoidectomy, were recruited. Main outcome measures: Diagnostic accuracy (i.e., comparison of RDO to HDO results) and test delivery time (i.e., days from consent signature to oximetry delivery) were recorded. Patient travel distances for HDO collection were calculated using home/delivery address postcodes and Google^®^ Maps data. Qualitative data were collected with two digital follow-up surveys. Results: All 32 patients that had both the HDO and RDO had identical oximetry results. The HDO mean delivery time was 87.7 days, while the RDO mean delivery time was 23.6 days (*p* value: <0.001). Qualitatively, 3/28 preferred the HDO, while 25/28 preferred the RDO (*n* = 28). Conclusions: The remote option is as accurate as the hospital option, strongly preferred by patients, more rapidly completed, and also an ideal investigation delivery method during certain emergencies, such as the COVID-19 pandemic.

## 1. Introduction

Snoring is a common symptom in the paediatric population. In children aged 2 to 14 years, it is estimated that 5–17% snore and that 3–5% within this age group suffer from obstructive sleep apnoea-hypopnea (OSA) [1,2,3,4]. OSA is characterised by partial or complete obstruction of the upper airways during sleep, with the disruption of normal ventilation and sleep patterns. Even mild untreated OSA is associated with adverse developmental, behavioural, metabolic, and cardiovascular outcomes in children [5,6].

Tonsillectomy and adenoidectomy are the most common surgical procedures in childhood, with over 12,000 procedures in Victoria, Australia, per year [7]. Currently, surgical removal of the tonsils (tonsillectomy) and/or adenoids (adenoidectomy) is recommended as the first-line therapy for most children with OSA. This treatment has an estimated treatment efficacy of resolving OSA in 73–95% in the paediatric population [8,9,10,11].

The proportion of respiratory complications in children undergoing general anaesthesia for elective surgery has been reported to range from 5 to 17% [12]. These complications may be minor (such as oxygen desaturations, cough, or minor obstructive events requiring repositioning) or major (such as laryngospasm, post-operative pulmonary oedema, airway obstruction, and respiratory failure, requiring non-invasive or invasive ventilation, and intensive care unit transfer). Children with OSA, however, are at increased risk for such early post-operative respiratory complications. A study by Lavin et al. found that post-operative respiratory complications occurred in 9.6% of non-obese, and 16.2% of obese children following adeno-tonsillectomy [13].

While polysomnography (i.e., formal in-laboratory level 1 sleep study) is the current gold standard in the diagnosis of OSA in children and, thus, in assessing the associated risks of post-operative complications, it is resource-intensive for both the patient and the health system, costly, time-consuming, and is generally associated with a very long waiting list [14]. Sleep oximetry (i.e., formal at-home level 4 sleep study), involving the single overnight home-based continuous recording of a child’s blood oxygenation via a cutaneous probe, on the other hand, is an easily accessible and reliable screening tool, with a high positive predictive value but low negative predictive value [4]. With the results of such pre-operative screening tools, the risk of serious respiratory complications can be mitigated, by placing higher-risk children in tertiary/quaternary care settings for their elective adeno-tonsillectomy, and allowing lower-risk children to safely undergo adeno-tonsillectomy in regional and rural, secondary care settings [15,16,17].

In Australia, pre-operative oximetry in children is typically limited to tertiary/quaternary paediatric centres. For example, in the state of Victoria, home oximetry screening for OSA is only provided by two quaternary centres in Melbourne. For paediatric patients attending one location, they must travel to this location near the city, undergo an instructional training session, use the oximeter overnight, ideally in the comfort of their own home, and return it in the morning. The same general procedure is true for the other location. However, this centre only offers a hospital-in-the-home oximetry service to patients residing in metropolitan Melbourne. In addition, parents of patients report that the waitlist can vary from weeks to several months.

Unfortunately for patients and families that live in regional centres, the collection of the overnight oximeter can have a significant impact on time and finances. It requires patients and their families to plan a trip to the nearest location (often 100 s of kms away), book a hotel or arrange other accommodation, and travel back home. This often includes the additional burden of time off work and school.

The COVID-19 pandemic further impacted all aspects of this overnight oximetry model. It made it increasingly difficult for regional patients to travel during lockdowns, while also increasing the importance of reducing unnecessary visits to healthcare locations. Further, the major public oximetry services in Victoria were reduced for periods of time during the pandemic, partly due to the impact of staff sick leave and partly due to the diversion of medical resources to other aspects of hospital care.

A novel home delivery oximetry kit, developed by Nebula Health, aims to circumvent many of these issues. It is a self-contained kit that can be mailed to patients throughout Australia, and contains a hospital-grade continuous pulse oximeter and sensors, an instructional video brochure, digital sleep diary, and written information. It is designed so that no additional training or coaching is required for the test to be completed by the patient, and can be mailed back to the reporting healthcare service the following day. In this way, the test can be completed in the familiar environment of the patient’s own home, and without the need for extensive travel or time away from work or school.

In this study, we sought to test the diagnostic accuracy and feasibility of delivery of this remotely-delivered oximetry (RDO) kit to rural and regional children, by comparing it against the current standard of care in home oximetry, the hospital-delivered oximetry (HDO) service.

## 2. Materials and Methods

Ethics approval was obtained through the relevant hospital research ethics committee.

### 2.1. Patient Recruitment

Over the period of (April 2019–April 2021), all consecutive paediatric patients (aged 2–18), diagnosed by a visiting Ear Nose and Throat Surgeon at a single rural hospital setting (Swan Hill, VIC, Australia), with OSA on history/examination, and booked for tonsillectomy +/− adenoidectomy, were recruited to the study. Each patient was referred for both the current standard in screening home oximetry, routine hospital-delivered oximetry (HDO), at “The City-Based Site”, and for the novel mail-out remotely-delivered oximetry (RDO) kit. Prior to study commencement, a pre-specified power calculation determined that a study size of at least 30 participants, each undergoing both the HDO and RDO tests, would allow for statistically significant results. The power-based sample size calculation assumed a conservative mean HDO delivery time of 30 days, a mean RDO delivery time of 25 days, a power of 80%, and a type 1 error of 0.05. Each patient served as their own control, thus allowing a direct comparison of the two techniques (HDO vs. RDO). Throughout the study, there was no direct cost to the patient or their family. Exclusion criteria included: neuromuscular disease, congenital syndromes, congenital heart diseases, chronic lung disease requiring oxygen therapy, previous trauma/burns to airway/face/neck, and/or weight <3rd or >95th percentile for age.

### 2.2. Remotely-Delivered Oximetry (RDO)

Following informed consent, a Research Assistant contacted the guardians to confirm kit delivery address and contact information. The RDO kit was then sent via regular registered domestic post. The kit comprised the pulse oximeter, sleep diary, written instructions, paid return shipping label, video brochure instructions, and two disposable finger/toe sensors. Automated instructions via text message and email, and an electronic version of the sleep diary were also simultaneously sent to the patients’ carers. The electronic instructions and sleep diary mimicked the hard-copy instructions in the oximetry kit.

The oximetry kit was received via postage, the oximetry study was performed without any direct training or instruction (beyond the written instructions and video brochure), and the kit and sleep diary were then returned using the self-addressed return packaging. Support was available for parents via phone during business hours. Upon receipt of the returned package, oximetry and sleep diary data were downloaded and collated using Profox^®^ software, the software was accessed throughout the study period (http://www.profox.net).

It was intended that patients complete the RDO study prior to the HDO study, in order to minimise any prior education, and thus reduce bias in performing the RDO.

### 2.3. Hospital-Delivered Oximetry (HDO)

As per the standard of oximetry care, children were first referred to The City-Based Site and an appointment for oximetry kit collection was arranged with the city location. On the appointment day, patients and guardians travelled to the city location and met with an allied healthcare worker, who explained how to use the oximeter. The patient and guardian took the kit to their arranged accommodation (for example, a hotel or housing provided by a non-profit organisation), performed the test overnight, recorded the sleep events in the paper-based sleep diary (i.e., waking, gasping, apnoeas, change of position, and waking for uresis), and returned the oximetry kit the following morning. The patient and guardian then travelled home, and a paediatric sleep physician then reported the study, following the methodology below.

### 2.4. Oximeter and Settings

Masimo Radical-7^®^ Co oximetry machines were used for both the RDO and HDO studies. The oximetry settings were identical for both the RDO and HDO studies. Continuous recording was employed, with an averaging time of 2 s and study recording resolution of 2 s. A single alarm was set to activate if there was finger/toe sensor signal loss overnight.

### 2.5. Reporting of Oximetry

To ensure consistent and objective evaluation of data, sleep physicians were unable to access the oximetry data of the patient’s corresponding oximetry test. Furthermore, each RDO and corresponding HDO test was reported by separate physicians to further ensure rigour. Therefore, each study was analysed by an independent, blinded paediatric sleep physician, who were trained in oximetry reporting as per accepted Australian reporting guidelines [18].

Standardised McGill oximetry scoring criteria [19] were used to generate the following possible results:

Normal study/inconclusive for OSA;

OSA, mild;

OSA, moderate;

OSA, severe.

If the oximetry study had insufficient data to report (<6 h), this was noted, and a repeat study was conducted.

### 2.6. Data Collection

The following information was collected for each patient: age, sex, test delivery time (measured as days from consent signed to oximetry kit delivery for both the HDO and RDO), oximetry result(s), and number of required repeats for the RDO tests. Patient travel distance for HDO collection was calculated using home address postcodes and Google^®^ Maps.

Two types of follow-up surveys were performed, one after the RDO test was performed, and another when both the RDO and HDO tests had been completed (see Table 1). Due to COVID-19, adjustments were made to the surveys (refer to “COVID-19 Adjustments” below for more detail).

Upon completion of both oximetry studies, the HDO results were compared with the RDO results.

### 2.7. COVID-19 Adjustments

The COVID-19 pandemic began to impact the study and required certain adjustments to the study protocol. When local COVID-19 cases were reported, rurally located patients’ guardians were reluctant to travel, and in particular, did not want exposure to a city healthcare site. At that time, a preliminary analysis was performed that demonstrated that all 32 patients that had both the HDO and RDO had clinically identical results. As the children involved had potential breathing disorders, they were considered high risk if exposed to COVID-19. For those reasons, the treating physician decided that the mandatory HDO was not in the patients’ best interest and an ethics amendment was submitted and approved. At this stage, the tracking of delivery times was discontinued because a legitimate comparison between the RDO and HDO delivery times was no longer possible. Otherwise, the study continued, analysing patient satisfaction scores, and slight edits were made to the surveys to reflect the optional HDO test; this included adding the questions in “Page 4” and “Page 5” of Survey 2 (see Table 1).

## 3. Results

### 3.1. Trial Participation

A total of 86 participants were consented to participate in the study. A total of 32 completed both the hospital-delivered oximetry (HDO) and remotely-delivered oximetry (RDO) (see Figure 1).

### 3.2. Patient Demographics

A total of 77 patients participated in the study, their demographics are represented in Table 2 (below).

### 3.3. Test Completion Times

The RDO had a mean delivery time of 23.6 days, while the HDO had a mean delivery time of 87.7 days with a *p* value: <0.001 (Figure 2).

### 3.4. Clinical Result Comparison

All 32 patients that had both the HDO and RDO had identical oximetry results (as shown in Figure 3). Thus, it can be concluded that the RDO was as accurate as the HDO in all patients who underwent both studies.

### 3.5. Repeat Rate

Out of a total of 77 tests, 7 remotely delivered oximetry (RDO) tests had to be repeated. The number of repeats for the hospital-delivered oximetry (HDO) was unknown. All RDO tests that were repeated were completed before the HDO test, without delaying the HDO test.

### 3.6. Patient-Reported Experience Measures

The majority of patients preferred the RDO over the HDO, this is shown in Figure 4.

### 3.7. Likelihood to Recommend Home Oximetry

For the question, “If a friend or relative needed a similar test, how likely are you to recommend (the home study) to them?” 1 (never) 2 3 4 5 (always).

The mean result was: 4.75/5.0, *n* = 64.

### 3.8. Thematic Analysis of Qualitative Data

Qualitative data were overwhelmingly positive in favour of the RDO. In fact, all feedback that expressed any preference towards the HDO is included in Table 3.

## 4. Discussion

### 4.1. Overview of Results

The effectiveness of home oximetry as a screening triaging device to mitigate respiratory risks in children undergoing elective adenotonsillectomy for OSA has already been demonstrated [15]. The disparity in access to timely paediatric adeno-tonsillectomy for children in Victoria based on rurality has also been demonstrated [7,8,9,10,11,12,13,14,15,16,17,18,19,20]. This study demonstrates the feasibility, accuracy, and patient preference for remotely delivered oximetry (RDO) in a real-world setting, thus potentially facilitating more timely and safer paediatric adeno-tonsillectomy to regional and rural children, and indeed, potentially for any child in Australia. 

### 4.2. Test Completion Times

Prior to the initiation of this study, the HDO wait times were reportedly approximately 3 months. Therefore, a primary goal of this study was to measure wait times, which are presented in Figure 2. The RDO had a significantly reduced wait time with a mean delivery time of 23.6 days, while the HDO had a mean delivery time of 87.7 days (*p* < 0.001). The reduced wait time (in the RDO group) has the potential to provide improved patient care through more rapid triaging for surgery and streamlined theatre lists. 

### 4.3. Clinical Equivalence and Test Quality

Seven of 77 RDO tests required repeat testing due to insufficient data collection. Of note, all RDO repeat studies were repeated before the HDO was performed, without needing to delay the HDO test. Therefore, in this context, even when the RDO does have to be repeated, it was still more time-efficient than the HDO option. It is also likely that the repeat rate could be decreased by adjusting the educational information provided to parents, and future options for remote oximetry data upload may also allow for immediate analysis of oximetry data and identification of an inadequate study, prior to return of the kit.

All patients performed the RDO study first and were, therefore, educated on how to perform oximetry prior to the HDO. The main purpose in performing the RDO and HDO tests in this order was to test how well the RDO option performed without any prior education from the HDO test. It also facilitated a more reliable measure of delivery times.

### 4.4. Qualitative Data

The patients’ guardians preferred the RDO over the HDO, with 25 out of 28 expressing preference for the RDO (Figure 4).

A thematic analysis of the qualitative data showed three main categories: comfort at home, convenience (financial and distance), and HDO preference (Table 3). The overwhelming amount of feedback was positive and focused on the comfort at home and convenience of the RDO.

The RDO was also performed by guardians without the formal training session required by the HDO, which is conducted by allied healthcare staff. In the context of COVID-19, and ongoing infection control protocols, this provides an extraordinary opportunity to extend care to regional and rural patients and reduce the substantial burden they face in performing oximetry. As technology rapidly progresses, this study also has wider implications for how other medical testing and care can be delivered remotely.

### 4.5. Limiting Factors and Improvements

On average, the delivery times for the RDO were longer than expected. This was due to the combination of a few factors:(1)A limited number of oximetry kits were provided by the Company (Nebula Health).(2)Delays in return of oximetry devices, partially attributed to lockdown restrictions, and other social circumstances.(3)Delays in reporting of received oximetry results. As this study progressed during COVID-19, the human resources of the respiratory/sleep physicians were re-deployed to more urgent COVID-19 care requirements, slightly delaying the research reporting of our studies.

Although the combination of the above factors resulted in delivery delays, these are all easily remedied with additional machines, personnel, and reporting physicians.

Another bottleneck in the process was the availability of sleep physicians for oximetry reporting. Of note, although the company provided the home oximetry kits, for reporting, sleep physicians were hired that had both experience in the reporting style of the HDO and were not being paid by the aforementioned company. However, due to a lack of available sleep physicians that fit the previous criteria, at times, the reporting delay was substantial and more physicians would need to be trained to perform this service on a large scale.

A substantial study limitation included a deviation from original protocol due to COVID-19 (discussed below).

In addition, whilst ideally every child would undergo a diagnostic level 1 sleep study to accurately confirm the presence/absence of OSA, the aim of this study was to compare the novel RDO to the established HDO OSA screening technique.

Furthermore, while this study was adequately powered to achieve a significant result, a larger cohort would allow a more robust comparison between delivery methods.

### 4.6. Less Resource-Intensive

A further benefit of the RDO vs. the HDO is that the RDO is less resource-intensive for the patient, carer, and likely also the healthcare system. For rural patients in this study undertaking HDO, they needed to travel an average of 392.99 km, while also booking accommodation and often taking time off work (Table 2). For healthcare systems, the amount of patient education required is likely less than the HDO, and oximeters can be stored in any location with access to a post office. However, there are ways in which the RDO system could become more expensive than the HDO. For example, during this study, there were phone calls from patients’ guardians occasionally asking questions about the test, which could theoretically translate to the need for greater personnel time and cost.

### 4.7. COVID-19 Adjustments

Although the advent of COVID-19 pandemic initially seemed to be a challenge, it resulted in a great learning opportunity. As local COVID-19 cases were reported, rural families were reluctant to travel into the city. In addition, various HDO appointments were delayed or cancelled altogether by the city-based hospital oximetry provider. This demonstrated the real-world utility of such a remote delivery service, which was able to continue in a COVID-safe manner, throughout the duration of the pandemic, with very little disruption to service, save for an incremental increase in postal delivery times.

This transitioned the study from a real-world study to one under additional stressors, and provided an opportunity to test the robustness of the RDO kit and processes. Interestingly, despite the challenges posed elsewhere by COVID-19, the pandemic did not meaningfully negatively impact the study. The only impact was that the HDO tests became optional due to delays and cancellations required by The City-Based Site. Otherwise, the study continued as planned, and participants continued to have high satisfaction with no discernible negative impacts. The success of the study, especially during the COVID-19 outbreak, demonstrates the efficiency and robustness of well-designed remote delivery care and has implications for other areas of point-of-care healthcare delivery.

## 5. Conclusions

This study demonstrated that remotely delivered oximetry (RDO) was:(1)Equivalent to hospital-delivered oximetry (HDO) in terms of accuracy;(2)Preferred by patients, with a high level of satisfaction;(3)More rapidly completed.

In this way, home-based oximetry provides a potentially scalable solution for the screening of children for obstructive sleep apnoea (OSA) undergoing adeno-tonsillectomy, thus improving regional and rural health equity. Based on this study, similar remote-care focused initiatives should be considered.

## Figures and Tables

**Figure 1 healthcare-11-00278-f001:**
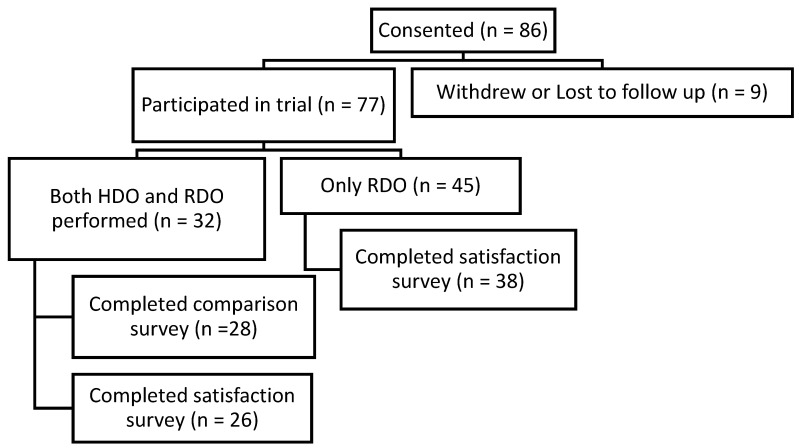
Trial participation: An overview of the participants in the trial; *n* = the number in each category.

**Figure 2 healthcare-11-00278-f002:**
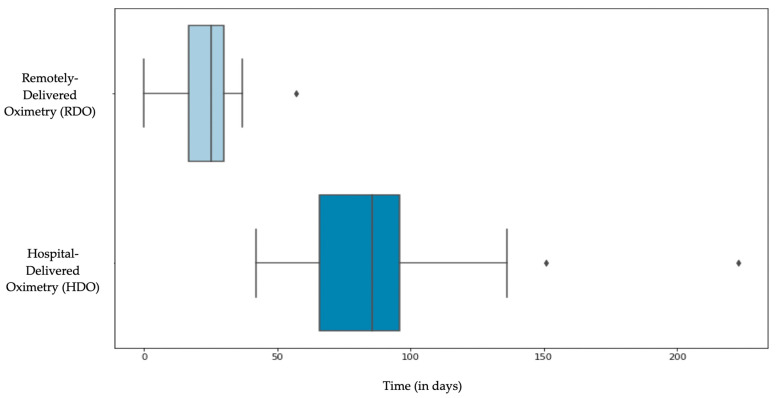
Remotely-delivered oximetry (RDO) vs. hospital-delivered oximetry (HDO) delivery times. The RDO had a mean delivery time of 23.6 days, while the HDO had a mean delivery time of 87.7 days (*p* value: <0.001). Delivery time was defined as the days from the date consent was signed to the date the oximetry kit was first in the patient’s possession.

**Figure 3 healthcare-11-00278-f003:**
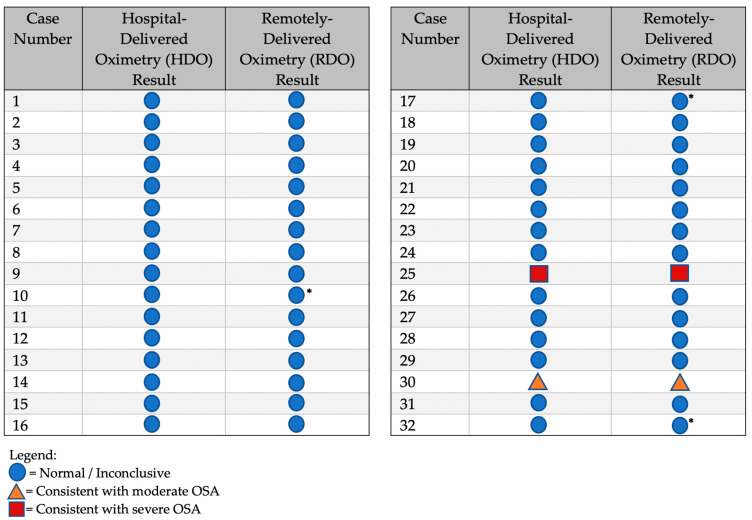
Clinical Result Comparison: The table compares the hospital-delivered oximetry (HDO) results to the remotely delivered oximetry (RDO) results. The 3 possible outcomes are shown alongside the corresponding result for the other test. Known tests that required a repeat are denoted with an asterisk. The repeat rate for city-based oximetry was unknown. ∗ = Known required repeats of RDO. Repeat rate of HDO was unknown.

**Figure 4 healthcare-11-00278-f004:**
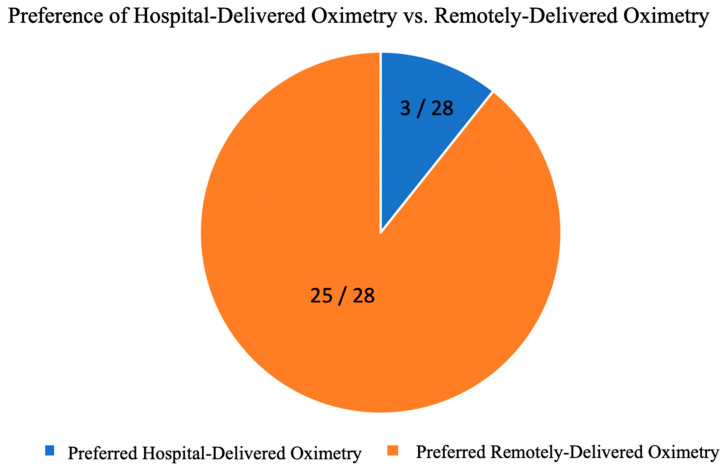
Preference for hospital-delivered oximetry (HDO) vs. remotely delivered oximetry (RDO). The graph shows that 3 out of 28 patients preferred the HDO and 25 out of 28 preferred the RDO.

**Table 1 healthcare-11-00278-t001:** Follow-up surveys: The follow-up surveys sent to patients during the trial are shown. Survey 1 was sent after the patient completed the RDO. Survey 2 was sent after the patient had completed both the RDO and HDO. However, as the COVID-19 pandemic progressed, appointments for the HDO were delayed or cancelled. Therefore, questions shown in “Page 4” and “Page 5” for Survey 2 were added, and Survey 2 was sent to patients regardless of completion of the HDO.

	Survey 1 (RDO Only Performed)	Survey 2 (RDO and HDO Performed)
**Page 1**	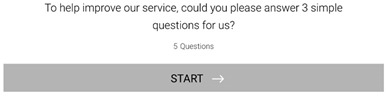	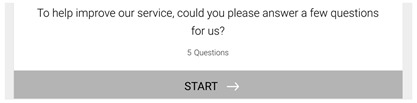
**Page 2**	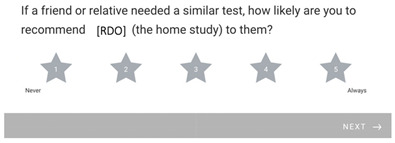	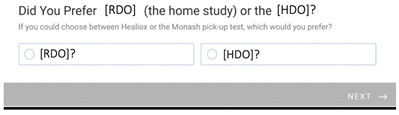
**Page 3**	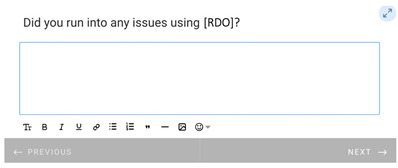	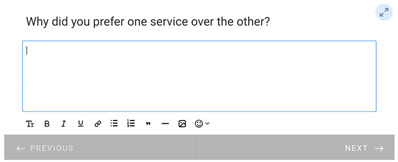
**Page 4**	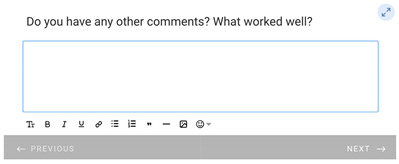	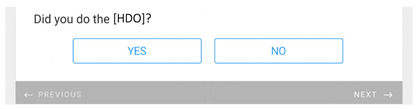
**Page 5**	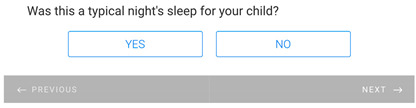	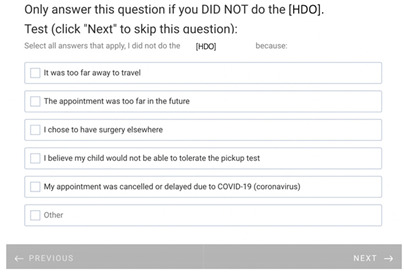 This question was only shown to respondents that answered “No” to the previous question
**Page 6**	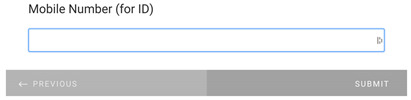	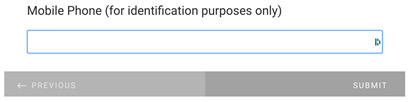
**Page 7**	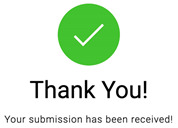	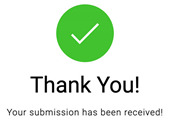

**Table 2 healthcare-11-00278-t002:** Demographics: The number of patients and their age, sex, and distance from the hospital-delivered oximetry (HDO) location are shown.

	All Patients	RDO and HDO Patients	RDO Alone
**Total (*n*)**	77	32	45
**Male (*n*)**	43	19	24
**Age at time of test (years)**	Mean = 5.49, SD = 2.71	Mean = 5.68, SD = 2.86	Mean = 5.34, SD = 2.61
**Patient distance to HDO (kms)**	Mean = 392.99, SD = 100.59	Mean = 367.19, SD = 108.77	Mean = 412.71, SD = 91.79

**Table 3 healthcare-11-00278-t003:** Thematic analysis of qualitative. The themes identified in the qualitative data included comfort at home, convenience (financial and distance), and HDO preference.

Theme	Quotations
Comfort at home	“I think you get a more accurate reading when they’re comfortable in their own home. The at home part was key! Would have been hard to travel with a toddler. Really fast turn around.” “100% the home one. Just easier. Didn’t have to travel. My kid was more relaxed because she was at home.” “Being regional having the sleep study machine sent to us was cheaper than traveling and [patient] was more comfortable at home.” “I had to travel with my child to [the city location] and stay in [non-profit accommodation] to do the test with [HDO]. It was essentially the same service except face to face for 10 min at [The City Based Site] to have the test explained to me. It would be more convenient and comfortable for my child if we could do the test at home. Because [HDO location] is a 4 h drive from home, so with driving I had to break it down in 2 days, drive up stay in a hotel, then stay at the [non-profit accommodation] then travel home. It’s a lot easier to do it in the comfort of your own home. The test can be done from the comfort of home, which is important for young children.” “I preferred the at home test as he slept a lot better in his own bed rather than at the hospital”
Convenience (travel and financial)	“The home study is a million times easier. Glad not to travel 550 kms and 100 s of $ to not travel to [the city location]” “So we didn’t have to drive 4.5 h and book a motel. Going to city affected our other children, had to bring 3 other kids [the city location]. The instructions were very clear.” “Answered in the previous survey. Home one was a lot simpler and didn’t cost us 100 s of dollars.” “I didn’t have to do an 8 h round trip with an overnight stay”
HDO preference	“Much easier to use” (in reference to the HDO test). “I preferred the [HDO] one because they explained it in more detail. Not just from a little book. It was a bit more info than in the mail one. But, if I were to do it again I would want the home one now that I know how to use it, but for the first time, the [The City Based Site] one explained it better. Having bullet points like: if this happens you should do this, would be better ([The City Based Site] one had that and the home one didn’t)” “I liked having someone explain it in person” “In some ways we preferred the [HDO] because [patient] was more at ease more relaxed in the [non-profit accommodation]. But the home one would be easier if we had to do it again.”

## Data Availability

Data available on request due to restrictions. The data presented in this study are available on request from the corresponding author. The data are not publicly available due to patient confidentiality requirements.

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
