# Peer review of "Demonstration of Accuracy and Feasibility of Remotely Delivered Oximetry: A Blinded, Controlled, Real-World Study of Regional/Rural Children with Obstructive Sleep Apnoea"

_healthcare, 2023, doi:10.3390/healthcare11020278_

Round 1

Reviewer 1 Report

Dear Authors, 

thank you for submitting the paper. Your study is well designed and written, it is very interesting that you faced the COVID-19 pandemic while this study was running. I read it with pleasure. 

The idea of auto-control group is great. I only suggest some better graphical presentation of the table (there are some technical mistake, as it is a screenshot). 

Author Response

Thank you very much for your kind words. We really appreciate your time and effort.

Yes, you are correct! We noticed a few of the tables and figures have had formatting issues that must have occurred during the submission process. We haven't attached fixes here, because we need to work with the editorial team about how to best submit/format them.

We have attached a revised manuscript incorporating the other reviewers comments as well.

Thank you again!

Reviewer 2 Report

C1. Please specify that HDO, which is a level 4 sleep test was used as the diagnostic test for detecting OSA. The reader is left wondering if level 1-2 polysomnography was performed.

C2. Please cite- Even mild untreated OSA is associated with adverse developmental, behavioral, metabolic, and cardiovascular outcomes in children on line 31

C3. Please explain your power calculation.

C4. Please enlist your study limitations. There was a significant deviation from original protocol (due to COVID-19), small sample size, level 4 sleep study used, limited data etc.  

Author Response

Thank you very much for time and thoughtful feedback! Your comments were very helpful.

Responses:

C1: Excellent point. We have included a few words and phrases throughout the paper to clarify this. These can be seen in lines 52, 56, 105, 313-315

C2: Great point! Citations included. Lines 36-37.

C3: Again, your comment was helpful. Edit on lines 109-111.

C4: Edits on lines 311-317

Thank you again. You clearly read and thought about the research carefully, which is sincerely appreciated!

Round 2

Reviewer 2 Report

The authors have addressed previously expressed concerns.